# High Prevalence of *Clostridioides difficile* Ribotype 176 in the University Hospital in Kosice

**DOI:** 10.3390/pathogens12030430

**Published:** 2023-03-08

**Authors:** Katarina Curova, Martin Novotny, Lubos Ambro, Anna Kamlarova, Viera Lovayova, Vladimir Hrabovsky, Leonard Siegfried, Pavol Jarcuska, Peter Jarcuska, Annamaria Toporova

**Affiliations:** 1Department of Medical and Clinical Microbiology, Faculty of Medicine, Pavol Jozef Šafárik University in Košice, Trieda SNP 1, 04011 Kosice, Slovakia; katarina.curova@upjs.sk (K.C.); viera.lovayova@upjs.sk (V.L.); vladimir.hrabovsky@upjs.sk (V.H.); leonard.siegfried@upjs.sk (L.S.); 2Department of Infectology and Travel Medicine, Faculty of Medicine, Pavol Jozef Šafárik University in Košice, Rastislavova 43, 04190 Kosice, Slovakia; martin.novotny@unlp.sk (M.N.); pavol.jarcuska@upjs.sk (P.J.); 3Center for Interdisciplinary Biosciences, Pavol Jozef Šafárik University in Košice, Technology and Innovation Park, Jesenna 5, 04001 Kosice, Slovakia; lubos.ambro@upjs.sk; 4Center for Clinical and Preclinical Research MediPark, Faculty of Medicine, Pavol Jozef Šafárik University in Košice, Trieda SNP 1, 04011 Kosice, Slovakia; anna.kamlarova@upjs.sk; 52nd Department of Internal Medicine, Faculty of Medicine, Pavol Jozef Šafárik University in Košice, Trieda SNP 1, 04011 Kosice, Slovakia; peter.jarcuska@upjs.sk

**Keywords:** *Clostridioides difficile*, *Clostridioides difficile* infection, antibiotics, toxin, ribotype

## Abstract

Dysbiosis of the gut microbiota, caused by antibiotics, plays a key role in the establishment of *Clostridioides difficile* CD). Toxin-producing strains are involved in the pathogenesis of *Clostridioides difficile* infection (CDI), one of the most common hospital-acquired infections. We cultured a total of 84 *C. difficile* isolates from stool samples of patients hospitalized at Louis Pasteur University Hospital in Kosice, Slovakia, that were suspected of CDI and further characterized by molecular methods. The presence of genes encoding toxin A, toxin B, and binary toxin was assessed by toxin-specific PCR. CD ribotypes were detected using capillary-based electrophoresis ribotyping. A total of 96.4% of CD isolates carried genes encoding toxins A and B, and 54.8% of them were positive for the binary toxin. PCR ribotyping showed the presence of three major ribotypes: RT 176 (n = 40, 47.6%); RT 001 (n = 23, 27.4%); and RT 014 (n = 7, 8.3%). Ribotype 176 predominated among clinical CD isolates in our hospital. The proportion of RT 176 and RT 001 in four hospital departments with the highest incidence of CDI cases was very specific, pointing to local CDI outbreaks. Based on our data, previous use of antibiotics represents a significant risk factor for the development of CDI in patients over 65 years of age.

## 1. Introduction

The human gut microbiota is a complex, dynamic, and heterogenous population of microorganisms, including bacteria, fungi, archae, and viruses. They interact with each other, as well as with the host, and significantly affect the host during homeostasis and disease. The number of microorganisms inhabiting the gastrointestinal tract has been estimated to exceed 10^14^, with the predominance of bacteria types belonging to the Firmicutes, Bacteroidetes, Actinobacteria, and Proteobacteria phyla [1,2,3]. The importance of microbiota for human health is enormous. In particular, the role of gut microbiota in shaping the host’s intestinal epithelium, production of beneficial compounds for the host, protection against pathogens, and regulation of host immunity has been extensively studied and proved [4,5,6,7].

External factors (antibiotics, dietary components, stress) and host factors can induce dysbiosis of the gut microbiota. Dysbiosis affects the structure and function of the gut microbiota, which leads to the selective enumeration of certain microbiota members, including pathobionts, and to dysregulated production of microbial products or metabolites potentially harmful to the host. As a result of the aforementioned pathological processes, a wide spectrum of diseases is being developed, in particular, irritable bowel syndrome, inflammatory bowel disease, colon cancer, or *Clostridioides difficile* infection, metabolic diseases (metabolic syndrome, obesity, and diabetes), autoimmune diseases (celiac disease, systemic lupus erythematosus, and others), allergies, and neural disorders (depression and anxiety) [3,8,9].

*Clostridioides difficile* is a spore-forming anaerobic Gram-positive bacillus, taxonomically belonging to the Firmicutes phylum. The CD is proved to be a part of normal gut microbiota; however, in very low amounts. Some strains of *C. difficile* can produce toxins that are involved in the pathogenesis of CDI [8,10]. Generally, toxigenic strains of *C. difficile* produce toxin A (enterotoxin, TcdA) and toxin B (cytotoxin, TcdB). In addition, 6–30% of *C. difficile* strains produce another toxin called binary toxin (*C. difficile* transferase, CDT). It is often found in so-called hypervirulent strains responsible for outbreaks of CDI in hospital settings [11,12]. Dysbiosis of the gut microbiota, predominantly by antibiotics (ATBs), plays a key role in the establishment of CD and also in the increased toxins secretion by a greater number of vegetative cells. Subsequent damage to the intestinal barrier stimulates a severe inflammatory response that leads to diarrhea. Symptoms of CDI range from mild to severe diarrhea, pseudomembranous colitis, toxic megacolon, bowel perforation, and sepsis [13,14,15].

Since the emergence of hypervirulent CD ribotype BI/NAP1/027 (RT 027) in 2000, the epidemiological situation of CDI has been changing to become one of the most common hospital-acquired infections [10]. The spread of hypervirulent strains in hospitals and healthcare settings has led to increased rates of morbidity, mortality, and medical costs worldwide. In addition to RT 027, other ribotypes also appear and circulate in Europe [16,17]. Therefore, a deeper molecular characterization of causative CD strains is required to identify a common RT cluster in a suspected CD outbreak or to monitor the emergence of new RTs. The aim of our study was to investigate gene-encoding toxins and ribotypes of CD isolates among patients hospitalized at Louis Pasteur University Hospital (LP UH) in Kosice. We also summarize the key risk factors of CDI, the distribution of CD ribotypes in hospital departments, and treatment strategies of CDI in our set of patients.

## 2. Materials and Methods

### 2.1. Study Design and Patients

Patients admitted to LP UH in Kosice between 1 January 2020 and 31 December 2020, who were at least 18 years of age at the time of admission, with suspicion of CDI and positive glutamate dehydrogenase (GDH) rapid test result, were included in this study. All stool samples of patients with suspicion of CDI were analyzed for the detection of GDH enzyme and toxins A/B by an immunochromatographic assay (Intermedical CLOSTRIDIUM TRIO TOXIN A/B/GDH) at the Department of Clinical Microbiology of LP UH in Kosice. Whereas the risk of acquiring infection with *C. difficile* is disproportionately higher in people aged over 65 compared to younger people, the group of patients older than 65 years was separated for the clinical characteristics and antibiotics used for treatment before onset of CDI.

### 2.2. Cultivation and Identification

Stool samples with positive GDH tests were submitted for *C. difficile* isolation via conventional culture methods. Each sample was pretreated using the alcohol-shock method. A total of 70% methyl alcohol (1 mL) was added to the stool sample (~1 mL, used directly), and the mixture was vortexed for 10 s every 15 min and incubated at room temperature for 45 min. Deposit (50–75 µL) was inoculated onto Brazier´s *Clostridium difficile* Selective agar (Oxoid). The inoculated plates were immediately transferred to the anaerobic workstation (Whitley A35, Don Whitley Scientific, Bingley, UK) and incubated at 36 °C for 48 h. The species identification of putative colonies was performed by MALDI TOF mass spectrometry (Bruker Daltonics, Billerica, MA, USA).

### 2.3. Isolation of Genomic DNA

*C. difficile* genomic DNA was extracted from the grown colonies on Brazier´s agar using the GRISP isolation kit (GRS Genomic DNA kit) in accordance with the manufacturer´s instructions. Extracted DNA samples were stored at −20 °C until used for PCR analysis.

### 2.4. Detection of Toxin Genes

All DNA samples were amplified for the 16S rDNA, *tcdA*, *tcdB*, *cdtA*, and *cdtB* genes of *C. difficile* in a single multiplex PCR, as described by Persson et al. [18]. The total PCR mixture final volume was 25 µL with 1.5 µL of genomic DNA, 12.5 µL of One Taq Master Mix (BioLabs), 4.75 µL of nuclease-free water, and 12 primers with the corresponding volume of each primer depending on its concentration. The amplification was carried out in Biometra thermal cycler (Biometra TOne 96 G, 230 V, Analytik Jena, Jena, Germany) according to the following protocol: one cycle of 30 s at 94 °C; 30 cycles of 30 s at 94 °C; 40 s at 53 °C; then 70 s at 68 °C; and the final extension of 5 min at 68 °C. The sizes of the amplification products were as follows: 1062 bp for the 16S-rRNA; 629 bp for the *tcdA*; 410 bp for the *tcdB*; 262 bp for the *cdtB;* and 221 bp for the *cdtA* gene. Amplification products were separated in 1.5% agarose gel in Tris-acetate-EDTA buffer, stained by ECO Safe (Uniscience), and visualized by UV light. The image was captured by Gel Doc TMEZ System (BIO-RAD) and analyzed using Image Lab Software (BIO-RAD).

### 2.5. PCR Ribotyping

PCR ribotyping was performed according to the standardized protocol for capillary-based electrophoresis of PCR ribotyping [19] with primers described by Bidet et al. [20]. These primers are complementary to the 3′ end of the 16S rRNA gene and the 5′ end of the 23S rRNA gene and allow amplification of the variable intergenic spacer region. The total PCR mixture consisted of 1 µL of genomic DNA, 12.5 µL of Hotstar Taq Master Mix (BioLabs), 10 µL of nuclease-free water, and 0.3 µL of each primer. The PCR protocol included one cycle of 30 s at 94 °C, 30 cycles of 20 s at 94 °C, 45 s at 52 °C, then 50 s at 68 °C, and the final cycle of 5 min at 68 °C. PCR fragments were analyzed in an ABI3130 automatic genetic analyzer (Thermo Fisher Scientific, St. Louis, MO, USA) in a 50 cm capillary loaded with Pop 7 polymer (Thermo Fisher Scientific). LIZ 1200 was used as the size standard (Thermo Fisher Scientific). Each sample length of fragments was determined using the GeneMapper v4.1 software (Thermo Fisher Scientific). The ribotyping profiles were compared with the Webribo database [21].

### 2.6. Antimicrobial Susceptibility Testing

The susceptibility of CD isolates to antimicrobials metronidazole (MTZ), vancomycin (VA), tigecycline (TGC), doxycycline (DO), tetracycline (TTC), and rifampicin (RIF) was tested by disc diffusion test on Wilkins–Chalgren agar (Oxoid). After the application of bacterial suspension (density 0.5 McFarland) and six antibiotic discs, the plates were incubated in the anaerobic workstation at 36 °C for 48 h. Inhibition zone diameters were evaluated using the BACMED 6iG2 automated AST reader and analyzer. Subsequently, BACMED 6iG2 automatically calculated the minimum inhibitory concentration (MIC) using an expert system. CD isolates were classified as susceptible or resistant to ATBs based on the breakpoints for interpretation of MIC issued by the European Committee on Antimicrobial Susceptibility Testing (EUCAST) or the Clinical and Laboratory Standards Institute (CLSI).

### 2.7. Statistical Analysis

For statistical comparison of the results, statistical methods of data processing and results evaluation were used. Processed research data were entered into the data tables and visualized in a graphical form (SPSS 21, GraphPad Prism 9). Statistical analysis by Wilcoxon test was used to determine whether there was a significant difference between the characteristics of patients (clinical characteristics, comorbidities, antibiotics, and the number of ATB classes) for the group of patients over 65 years of age. All *p*-values of less than 0.05 were considered statistically significant.

## 3. Results

### 3.1. Characteristics of Patients and C.difficile Isolates

From January 2020 to December 2020, GDH positivity was confirmed in 108 stool samples from 102 patients. After discarding 24 samples due to duplicity, lack of material, and negative culture, a total of 84 *C. difficile* isolates taken from hospitalized patients suspected of CDI were recovered and further characterized. From the total number of 84 CD isolates, 40 (47.6%) were obtained from females and 44 (52.4%) from males. The age of patients ranged from 31 to 98, and 70 (83.3%) patients were older than 65. As it comes to the origin of CDI, most of the cases—75 (89.3%) were hospital-acquired CDI (HA CDI), 4 (4.8%) cases were community-acquired CDI (CA CDI), 3 (3.6%) cases were recurrent CDI, and in 2 (2.4%) cases the origin of CDI was unknown. A total of 17 (20.2%) patients died. In patients with HA CDI, hospitalization in the same department was recorded in 42 cases, previous hospitalization in the same hospital during the last 4 weeks in 32 cases, and hospitalization in another hospital during the last 4 weeks in 2 cases as the origin of CDI. The clinical characteristics of patients included in this study are presented in Table 1. Statistical analysis did not confirm the significance between the clinical characteristics of patients for the group of patients over 65 years of age and also between comorbidities for the group over 65 years of age. In the same way, no statistical differences were confirmed for the group of patients under 65 years of age.

A total of 68 CDI patients (80.9%) were treated with antibiotics (ATBs) within three months before the episode of CDI, while 49 (58.3%) patients used two and more ATBs belonging to different classes within the same period. The most commonly used ATBs were cephalosporins (n = 41, 8.8%), fluoroquinolones (n = 23, 27.4%), clindamycin (n = 18, 21.4%), trimetoprim/sulphametoxazole (n = 11, 13.1%), and macrolides (n = 10, 11.9%). Details about the types and number of ATB classes used by patients are summarized in Table 2. Statistical analysis confirmed a statistically significant difference between the group of antibiotics used most frequently before the onset of CDI (cephalosporins, fluoroquinolones, clindamycin, trimetoprim/sulphametoxazole, macrolides, carbapenems) for the group of patients over 65 years of age. Significance was demonstrated at the 95% level of significance (*p* < 0.0029). On the other hand, the significance between the group of antibiotics less frequently used before the onset of CDI (aminoglycosides, metronidazole, amoxicillin/clavulanic acid) for the group over 65 years of age and also between the number of ATB classes for the group over 65 years of age was not confirmed by statistical analysis. No statistical differences were confirmed for the group of patients under 65 years of age.

Distribution of CD isolates per hospital department was as follows: 20 cases from the 4th Department of Internal Medicine (providing comprehensive curative-preventive care within internal medicine); 18 cases from Infectology; 17 cases from Traumatology; 12 cases from the 1st Department of Internal Medicine (profiling in gastroenterology); 3 cases from Neurosurgery; 3 cases from the 1st Department of Surgery; 3 cases from Pneumology; 2 cases from the 2nd Department of Surgery; 2 cases from Hematology; 1 case from Urology; 1 case from Psychiatry; 1 case from Orthopedics; and 1 case from Department of Anesthesiology and Intensive Medicine.

For the ATB treatment of CDI, vancomycin was used in 49 cases, both vancomycin and metronidazole—in 22 cases, metronidazole in 7 cases, and fidaxomicin in one case. Fecal microbiota transplantation (FMT) was used in addition to ATBs in two cases. To clarify, fidaxomicin and FMT were applied to treat recurrent CDI. In three cases, there was no antibiotic treatment due to an asymptomatic course, and in two cases, treatment was unknown.

### 3.2. Toxins of C. difficile Isolates

Among the 84 stool samples selected for the study, the analysis of toxins A/B by rapid test revealed 69 (82.1%) positive stool samples. The presence or absence of toxin genes was determined by multiplex PCR. In total, 81 (96.4%) CD isolates carried genes *tcdA* and *tcdB* encoding toxins A and B. A total of 46 of them (54.8%) also carried genes *cdtA* and *cdtB* encoding binary toxin. Three different toxin gene profiles were identified among the *C. difficile* isolates: 46 (54.8%) had *tcdA^+^ tcdB^+^ cdtA^+^ cdtB*^+^ genotype; 35 (41.6%) had *tcdA^+^ tcdB^+^ cdtA^−^ cdtB*^−^ genotype; and three (3.6%) isolates were non-toxigenic (*tcdA^−^ tcdB^−^ cdtA^−^ cdtB^−^* genotype). In this study, 12 (14.3%) toxigenic *C. difficile* isolates were identified from stool samples previously evaluated by the rapid test as toxin A/B negative; 7 had *tcdA^+^ tcdB^+^ cdtA^+^ cdtB^+^* genotype, and 5 had *tcdA^+^ tcdB^+^ cdtA^−^ cdtB^−^* genotype.

### 3.3. Ribotypes of C. difficile Isolates

By PCR ribotyping, three ribotypes were detected with the highest frequency: RT 176 (n = 40, 47.6%); RT 001 (n = 23, 27.4%); and RT 014 (n = 7, 8.3%). The remaining ribotypes (n = 12) were RT 020, RT 010, RT 027, RT 078, RT 003, and new RTs. They did not exceed three isolates per RT profile. In two cases, ribotyping failed (Figure 1). Table 3 indicates the association of CD ribotypes with three toxin gene profiles identified among the *C. difficile* isolates. The highest incidence of CDI and frequently isolated RTs was observed in four hospital departments (Figure 2). The isolates of RT 176 were most common in the fourth Department of Internal Medicine (n = 15), Infectology (n = 10), and the first Department of Internal Medicine (n = 6), and the isolates of RT 001 were most common in Traumatology (n = 15).

### 3.4. Antimicrobial Resistance of C. difficile Isolates

Antimicrobial susceptibility testing of CD isolates to six antibiotics and evaluation using the BACMED 6iG2 revealed 31 (36.9%) isolates susceptible to all tested ATBs. The resistance to RIF was observed most frequently in 47 out of 84 CD isolates (56%). Among these isolates, the majority (n = 40) belonged to RT 176. Five isolates (6%) were resistant to VA, and four isolates (4.8%) to TTC and DO. Three (3.6%) isolates were resistant to TGC, and one isolate (1.2%) to MTZ.

## 4. Discussion

*C. difficile* is one of the most frequently reported nosocomial microorganisms. Several risk factors contribute to the development of CDI, including the use of antibiotics, proton pump inhibitors, different comorbidities (arterial hypertension, diabetes mellitus, malignancies, chronic kidney diseases, COVID-19), older age, and long-term hospitalization [22,23]. In our cohort, we recorded 80.3% of patients aged over 65. The use of antibiotics within 3 months before the onset of CDI was confirmed among 81.4% of patients, and taking proton pump inhibitors during the mentioned time period among 50% of patients. The most common comorbidities were arterial hypertension (66.6%), diabetes mellitus (22.6%), oncological diseases (15.4%), and COVID-19 (10.7%). Similar findings have been reported by other studies [23,24,25,26].

The most significant influence on the incidence of CDI in hospital settings is the use of two and more classes of antibiotics, which alter the gut microbiota and are responsible for the physiological protection of the gastrointestinal tract against colonization by pathogens, including CD [27]. In our cohort, the use of two and more classes of ATB was detected in 58.3% of patients. Some ATBs, such as cephalosporins, clindamycin, and, more recently, fluoroquinolones, are known to carry a higher risk of CDI than others. In particular, cephalosporins have emerged as the ATBs with the highest relative and attributable risk of CDI [27]. During the COVID-19 pandemic, the number of CDI patients receiving high-dose antibiotic therapy with predispositions to antibiotic-associated diarrhea and the development of CDI has increased rapidly in some countries [25]. In our study, the most frequently used ATBs were cephalosporins (48.8%), fluoroquinolones (27.3%), clindamycin (21.4%), trimetrophim/sulphamethoxazole (13.1%), and macrolides (11.9%). Amoxicillin/clavulanic acid (7.1%) was a less frequent ATB. In addition, statistical analysis confirmed the existence of a statistically significant risk of ATB use in the group of patients aged over 65. Significance was demonstrated at the 95% level of significance (*p* < 0.0029). Antibiotic policy within individual countries and regions is very specific and different, which can be subsequently observed in relation to CDI. According to a study from Israel [28], 48.8% of patients were treated with cephalosporins and 6% with macrolides. In the study summarizing the ATB use in 9 hospitals from 7 European countries [29], fluoroquinolones were used by 24.4% of patients. In the study from Korea [30], clindamycin was used by 13.61% and trimetrophim/sulphamethoxazole by 19.78% of patients. The usage frequency of amoxicillin/clavulanic acid ranges from 0.19% to 46.9%, as demonstrated by several studies [25,29,31,32,33].

Currently, the antibiotics vancomycin and fidaxomicin are the first-line antibiotics for the treatment of CDI. Metronidazole should only be used in mild to moderate disease courses among younger patients who have no or few risk factors for recurrence. The treatment success rate among these patients is up to 89% [22,34,35]. Fidaxomicin has a high efficacy against CD without a significant negative effect on the gut microbiota. Czepiel et al. [10] indicate that the efficacy of fidaxomicin is comparable to vancomycin, and in some groups, it is even more effective in reducing CDI recurrence. In the case of recurrent CDI, fecal microbiota transplantation is the second line of treatment [36], with a success rate of 80–92% among patients suffering from recurrent and primary CDI [36,37]. In our study, 49% of patients were treated with vancomycin, 22% with vancomycin/metronidazole, and 7% with metronidazole. Fidaxomicin was used in one and FMT in two of the three cases of recurrent infections during the year 2020. In these three patients, the treatment was successful, and no further recurrences occurred.

Antibiotic misuse and the emergence of new hypervirulent *C. difficile* strains, mainly in nosocomial environments, led to a global increase in CDI incidence rate [15,16]. In Europe, the hypervirulent epidemic RT 027 strain was first reported in England in 2005 and has rapidly spread to other European countries [38]. Other ribotypes, such as RT 176, are genetically and phenotypically similar to RT 027 [39]. CDI caused by these RTs is associated with local outbreaks and a higher rate of complications and recurrences [40]. A number of other RTs are also emerging and circulating in Europe, namely, RT 001/072, RT 014/020, RT 140, RT 002, RT 010, and RT 078 [16,17]. In our study, RT 176 was confirmed in 40 CD (47.6%) isolates, representing the most frequently occurring ribotype in the total set of 84 CD isolates. This RT was also dominant among 46 toxigenic CD isolates carrying gene-encoding toxin A, toxin B, and binary toxin. On the other hand, RT 027 was detected in only one isolate (1.2%).

RT 176 is common in Eastern European countries (Czech Republic, Poland, Hungary, Slovakia) [39] with the varying incidence among hospitals, regions, and countries. Several studies have already investigated the prevalence of hypervirulent CD strains in Slovakia. A 5-month study carried out by Krehelova et al. [41] in three eastern Slovakian hospitals in 2017 found RT 176 in 3 out of 66 CD isolates (4.5%) recovered from GDH-positive stool samples. The LP UH is in the same region, but the incidence of RT 176 is higher. Novakova et al. [42] provided a 6-month study in a tertiary-care center in the northern part of Slovakia between January and July 2016. During this period, she examined 114 toxin-A/B-positive stool samples. RT 176 was detected in 75 (65.8%) CD isolates and was found to be the cause of CDI outbreaks among patients. In a national CDI surveillance in Slovakia (October 2016–December 2016), a total of 78 CD isolates from 12 hospitals were analyzed. The incidence of RT 176 was 29.5% (n = 23) [43]. In the Czech Republic, Krutova et al. reported a 26.7 % incidence of RT 176 for the period of 2013–2015 [39] and 11.6% for October–December 2017 [44]. Beran et al. [45] reported an incidence of 60.9% in 2011–2012, and Kracik et al. [46] 31.3% in 2017–2018. The incidence of RT 176 at the level of 6.7% was confirmed in Hungary [47], while in Poland, it was 2.8% [48]. RT 027 probably occurs only rarely in Slovakia. This is indicated by the results of our study, as well as the study of Novakova [43], reporting 1.3%, and Krehelova [41], reporting 0% incidence. RT 027 is also rare in the neighboring Czech Republic [39,44,45,46]. The predominance of RT 027 in recent years has been noted in Poland and Hungary, with the incidence of 82.4% and 45.8%, respectively [47,48].

The second and third most prevalent ribotypes in this study, RT 001 (27.3%) and RT 014 (8.3%), respectively, are characterized by the production of toxins A and B. Both RTs were already known to be circulating in European countries, including Slovakia, reporting 56.1% of RT 001 [41], 59% of RT 001 [43], the Czech Republic with 24% and 33.5% of RT 001 [39,44], 9.4% and 13% of RT 014 [45,49], Hungary with 11.4% of RT 014 [47], and Germany with 22.2% of RT 014 [50].

The proportion of individual ribotypes in the four departments of LP UH with the highest incidence of CDI cases was very specific and pointed to local outbreaks of CDI. RT 176 was the main ribotype that caused four local outbreaks in the fourth Department of Internal Medicine, three outbreaks in the Infectology, and two in the first Department of Internal Medicine. RT 001 was responsible for three local outbreaks in Traumatology and was found to be predominant in this department. Finally, ribotype 014 was also associated with local outbreaks in two departments (Infectology and Hematology). Transfers of patients among all these departments were observed and contributed to the emergence of new infections. Our results indicate that the proportions of CD ribotypes are probably highly connected to the specific hospital departments, and the nosocomial transmission of CD strains from one department to another is common. Similar findings are stated by several studies [46,51,52]. Most of these outbreaks were recorded between August and December, during the second wave of the COVID-19 pandemic. The situation in Slovak healthcare during this period was very problematic. In hospitals, including LP UH, there were not enough beds for patients with COVID-19 and other diseases, and the medical staff was exhausted and undersized. Strict antibiotic stewardship programs aimed to reduce the use of broad-spectrum antibiotics (third-generation cephalosporines, fluoroquinolones, macrolides, and others) were not followed. In contrast, ATB prescription and use were disproportionately high. The health condition of many patients, especially those over the age of 65, requires prolonged hospitalization. In the field of prevention and control of the spread of CDI, a team of experts was established in LP UH even before the outbreak of the COVID-19 pandemic. This team was very active in CDI surveillance and efforts to implement and manage preventive measures, including personal protective equipment, patient isolation, visit restrictions, and continuous education of the medical and cleaning staff. From the summer of 2020, it was not possible to continue the mentioned activities due to the worsening situation in the hospital. Not only the SARS CoV-2 virus but also *C. difficile* spread rapidly in hospital departments. The situation finally stabilized, and team activities resumed in 2022.

In recent years, an increasing number of CD isolates resistant to antibiotics has been observed [27]. In our cohort, results of antimicrobial susceptibility testing point to 36.9% of CD isolates susceptible to all tested ATBs. A total of 56% of CD isolates were resistant to rifampicin. Between July 2011 and July 2014, CD’s resistance to rifampicin was notable in Hungary, Italy, and the Czech Republic, with percentage values of 38.7–56.6%, 36.6–47.0%, and 40.0–64%, respectively [53]. Sholeh et al. [54], in a systematic review, described 42.3% resistant isolates. Aptekorz et al. [48] found 40.5% resistant isolates in 13 hospitals in Poland, with over 90% of them belonging to RT 027 and 33.3% to RT 176. Among our isolates, all isolates belonging to RT 176 (n = 40) exhibited rifampicin resistance. Vancomycin resistance in 6% and metronidazole resistance in 1.2% of CD isolates was found in this study. Vancomycin resistance is higher compared to the CDI surveillance of 2016 in Slovakia [43], where no reduced susceptibility to vancomycin was observed. While Freeman et al. [53] reported rare resistance to vancomycin (0.1%) and metronidazole (0.2%) in their study, Sholeh et al. [54] reported higher vancomycin and metronidazole resistance (3.7% and 3.2%). Finally, 4.8% of CD isolates were resistant to tetracycline and doxycycline, and 3.6% of isolates to tigecycline. Our results are comparable to the results of Sholeh et al. [54] for tigecycline (1.6%) but not for tetracycline (18.2%). The resistance of CD isolates to ATBs may create a survival advantage for resistant strains causing therapeutic failure and increasing chances of recurrence. This effect we observed in three of five CD isolates exhibiting vancomycin resistance (death in one case and recurrence in two cases). It is considered that CD’s resistance to ATBs can be multifactorial, and the need for thorough monitoring of clinical CD isolates is very high as it could help to identify new genotypic and phenotypic characteristics.

## 5. Conclusions

In conclusion, this is the first study of *C. difficile* isolates in the Louis Pasteur University Hospital in Kosice, Slovakia, which assessed complete cultivation and molecular biology-based characterization of *C. difficile* isolated from hospitalized patients over a period of one year. It is clear that previous consumption of antibiotics, especially cephalosporins, fluoroquinolones, and clindamycin, represents a significant risk factor for the development of CDI among patients aged over 65 years. This finding indicates a need for better management of antibiotic treatment, focused on careful selection and/or reduction of ATB prescriptions. CD genotype *tcdA^+^ tcdB^+^ cdtA^+^ cdtB^+^* and the ribotype 176 characterized by rifampicin resistance predominated among clinical *C. difficile* isolates. The occurrence of RT 176 and RT 001 in selected departments and transmission among departments points to the epidemic situation. This underlines the need for continuous testing of *C. difficile* and the setting of effective strategies to prevent CDI and its further spread.

## Figures and Tables

**Figure 1 pathogens-12-00430-f001:**
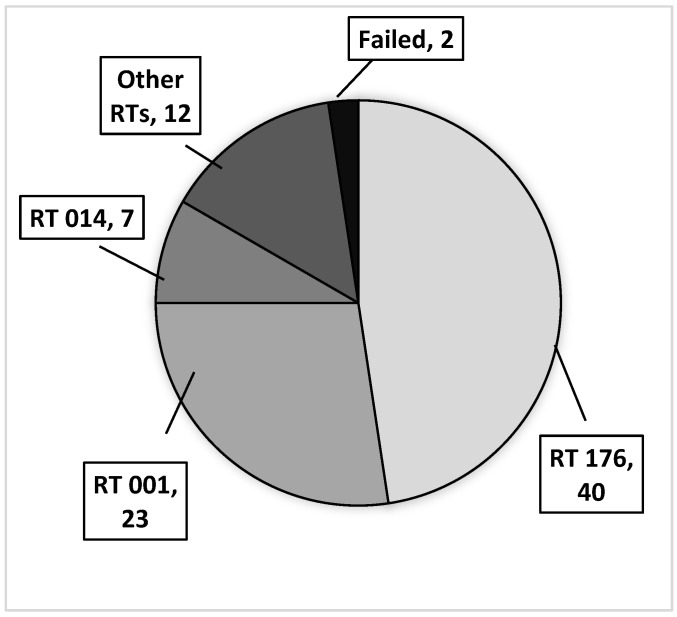
Frequency of PCR ribotypes in 84 *C. difficile* isolates.

**Figure 2 pathogens-12-00430-f002:**
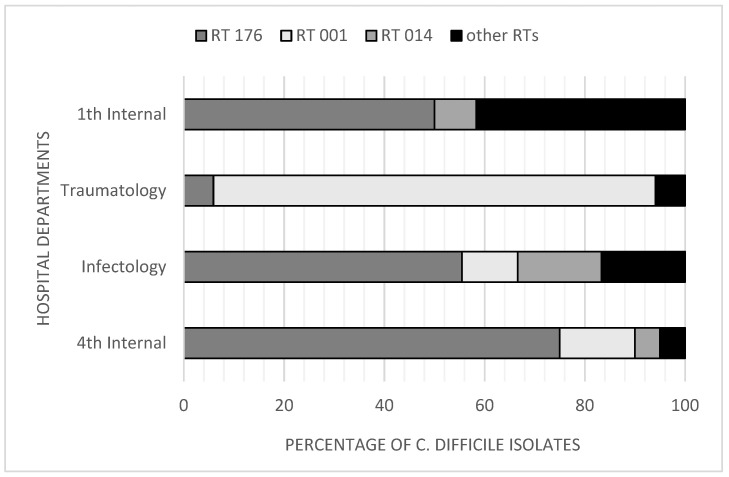
Distribution of *C. difficile* ribotypes in four hospital departments with the highest incidence of *C. difficile* infections. 1st Internal, 1st Department of Internal Medicine; 4th Internal, 4th Department of Internal Medicine.

**Table 1 pathogens-12-00430-t001:** Clinical characteristics of patients included in the study.

Clinical Characteristics	Overall (n = 84) No (%)	Age Over 65 (n = 70) No (%)	*p*-Value
Hospital-acquired CDI	75 (89.3)	62 (88.6)	
Death	17 (20.2)	16 (22.9)	
Previous use of ATBs *	68 (80.9)	57 (81.4)	n.s.
Previous use of PPIs *	42 (50)	34 (48.6)	
**Comorbities**			

Hypertension	56 (66.6)	49 (70)	
Diabetes mellitus	19 (22.6)	17 (24.3)	
Malignancy	13 (15.4)	12 (17.1)	n.s.
COVID-19	9 (10.7)	7 (10)	

* Medication within three months before onset of CDI; CDI, *Clostridioides difficile* infections; ATBs, antibiotics; PPIs, proton pump inhibitors; Statistical significance between the characteristics of patients (clinical characteristics and comorbidities) for the group Age over 65: *p* ˂ 0.05 *, n.s., not significant.

**Table 2 pathogens-12-00430-t002:** Summary of antibiotics used for treatment of patients before onset of CDI.

Antibiotics *	Overall (n = 84) No (%)	Age Over 65 (n = 70) No (%)	*p*-Value
Cephalosporins	41 (48.8)	38 (54.3)	0.0029 **
Fluoroquinolones	23 (27.3)	22 (31.4)	
Clindamycin	18 (21.4)	17 (24.3)	
Trimethoprim/Sulphametoxazole	11 (13.1)	9 (12.9)	
Macrolides	10 (11.9)	7 (10)	
Carbapenems	8 (9.5)	5 (7.1)	
Aminoglycosides	7 (8.3)	6 (8.6)	
Metronidazole	6 (7.1)	6 (8.6)	n.s.
Amoxicillin/Clavulanic Acid	6 (7.1)	4 (5.7)	
**Number of ATB classes ***			

1 ATB	19 (22.6)	17 (24.3)	
2 and more ATBs	49 (58.3)	40 (57.1)	n.s.
No ATBs	14 (16.7)	11 (15.7)	
Unknown	2 (2.4)	2 (2.9)	

* Medication within three months before onset of CDI; ATB, antibiotic; Statistical significance between the characteristics of patients (antibiotics and number of ATB classes) for the group aged over 65: *p* ˂ 0.05 *, *p* ˂ 0.01 **, n.s., not significant.

**Table 3 pathogens-12-00430-t003:** Toxin gene profiles and ribotypes of the 84 *C. difficile* isolates.

Ribotype	No of Isolates		Toxin Gene Profile	
		*tcdA^+^ tcdB^+^ cdtA^+^ cdtB^+^*	*tcdA^+^ tcdB^+^ cdtA^−^ cdtB^−^*	*tcdA^−^ tcdB^−^ cdtA^−^ cdtB^−^*
RT 176	40	40		
RT 001	23		23	
RT 014	7		7	
RT 078	2	2		
RT 020	2		2	
RT 027	1	1		
RT 003	1		1	
RT 010	1			1
New	5	2	1	2
Failed	2	1	1	
Total	84	46	35	3

## Data Availability

The datasets generated and/or analyzed during the current study are available from the corresponding author upon reasonable request.

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
