# Peer review of "High Prevalence of Clostridioides difficile Ribotype 176 in the University Hospital in Kosice"

_pathogens, 2023, doi:10.3390/pathogens12030430_

Round 1

Reviewer 1 Report

You have put a good effort to complete the study.  The paper is well written, but this can be improved better. I have the following suggestions. 

Line 92 – Was the stool sample used directly or is this 1 ml of a diluted stool sample?

Lines 110-111 – Provide the product size for each gene.

Line 175-185 – Please insert a table to show the RTs associated with three toxin gene profiles

Lines 308-309: This sentence talks about the genotype tcdA+ tcdB+ cdtA+ cdtB+ and at the end talks again about binary toxins.  Positive for cdA and cdB means the presence of binary toxins.

I think the future directions should be expanded and antibiotic susceptibility is also important here. The study has confirmed antibiotics as a risk factor. Have you tested the isolates collected from those individuals who were on antibiotic treatments for antimicrobial susceptibility? Were those strains antibiotic resistant? 

Author Response

Dear reviewer,

I want to thank for the highly professional revision, comments and suggestions. The changes are incorporated in the manuscript.

  • Line 95 – stool sample was used directly
  • Lines 113-115 – sizes of PCR products are provided
  • Line 211 - Table 3 (Toxin gene profiles and ribotypes of the 84 C.difficile isolates)is inserted
  • Line 338 – sentence is corrected
  • ATB susceptibility testing is realized by disc diffusion method and evaluated by automatic analyzer BACMED that recalculates inhibition zone to MIC value. Our results are not included as the method is not yet recommended by EUCAST. Briefly these results: 56% of islates resistant to rifampicin, 1.2% resist.to metronidazole, 6% to vancomycin, 3.6% to tigecyclin, 4.8% to doxycyclin

Reviewer 2 Report

The manuscript provides sound information on the prevalence of particular Clostridioides difficile types in one hospital and its departments. To my opinion, it would be interesting to summarize phenotypical, genotypical and patient information including antibiotic - and proton pump inhibitor use  as well as disease status, in one or more tables or preferentially a principal component analysis plot. This would show in one or two views the distribution of various features. 

Please add to line 138 to 142 where the distribution of CDI origin is coming from. It is presented as a result without a presenting data. 

Author Response

Dear reviewer,

I want to thank for the highly professional revision, comments and suggestions.

Line 147-151 – the origin of CDI is provided.

The idea to summarize all these data in table/analysis plot is interesting. For this publication, it is not possible due to further adjustments. However, we will keep this in mind when preparing further publication.

Sincerely

Katarina Curova

Reviewer 3 Report

Dear authors, 

thank you for this manuscript which interestingly identified the proportions of CD ribotypes among your hospital. The study was conducted with a good scientifically sound. The research methodology is sufficiently described. The discussion is thorough, concise, and includes all current data available on this topic. The conclusions are in accordance with the results. However minor points should be discussed : 

-        The authors should justify in the methods section the interest of systematically separating the observations for patients over 65 years-old. This was done in particular in Table 1 and Table 2.

-        Lines 153 -157 in results: Tables 1 and 2 are informative and descriptive statistics that help the reader knowing which antibiotics were used. However, an additional table is suggested for the two-way anova results as it is more conclusive to know which parameter has shown a statistically significant difference on the population. 

-        Line 162-164: What do the different internal medicine departments refer to (gastroenterology ?). What medical specialties are practiced there? Please specify or precise which art of medicine is practiced in these units (1st and 4th department of internal medicine). 

-        Figure 1: Please identify the ribotypes on the "circle" as the shades of grey are difficult for the reader to identify.

-        Figure 2 : The interest of showing the proportions of the different ribotypes according to the different units is weak insofar as an epidemic necessarily increases the same ribotype within a unit. It is therefore difficult to conclude with certainty that the proportions of ribotypes are highly connected and specific to a unit when an infection prevention and control problem is probably at the root of these epidemics. How do the authors defend these arguments other than on the basis of observed ribotype proportions? Are hygiene measures scrupulously respected in the hospital ? What hygiene precautions are put in place when a C. difficile patient is isolated ? Is there any team managing the infection prevention and control ? This should be discussed or at least mentioned.

-        The conclusions state on the need for a better antibiotic management which is probably one of the many root cause explanations for C. difficile hospital outbreaks. Is there any antimicrobial stewardship team or pharmaceutical team active for this surveillance in your hospital ? Or any infection prevention and control programs/teams? 

-        Finally, the term “comorbidities” should be adapted throughout the manuscript. 

Author Response

Dear reviewer,

I want to thank for the highly professional revisions, comments and suggestions. The changes are incorporated in the manuscript.

Line 88-91 – The interest of separating our observations for patients over 65 years-old is justified

Lines 150 and 166 – Tables are corrected, statistics is included

Line 207 – figure 1 is corrected

Lines 314-328 - the situation regarding the occurrence of ribotypes in the departments, hygiene measures, existence of team managing CDI prevention and control, ATB stewardship and other problems is explained (COVID 19 pandemic, collapsing healthcare, overcrowded hospitals). 

Sincerely

Katarina Curova